# Evolution of substrate specificity in a retained enzyme driven by gene loss

**Ana Lilia Juárez-Vázquez[1], Janaka N Edirisinghe[2,3], Ernesto A Verduzco-Castro[1], Karolina Michalska[4,5], Chenggang Wu[6], Lianet Noda-García[1†], Gyorgy Babnigg[4], Michael Endres[4], Sofía Medina-Ruíz[1‡], Julián Santoyo-Flores[7], Mauricio Carrillo-Tripp[7§], Hung Ton-That[6], Andrzej Joachimiak[4,6,8], Christopher S Henry[2,3], Francisco Barona-Gómez[1]\***

[1]Evolution of Metabolic Diversity Laboratory, Unidad de Genómica Avanzada (Langebio), Cinvestav-IPN, Irapuato, Mexico; [2]Computing, Environment and Life Sciences Directorate, Argonne National Laboratory, Lemont, United States; [3]Computation Institute, University of Chicago, Chicago; [4]Midwest Center for Structural Genomics, Biosciences Division, Argonne National Laboratory, Lemont, United States; [5]Structural Biology Center, Biosciences Division, Argonne National Laboratory, Lemont, United States; [6]Department of Microbiology and Molecular Genetics, University of Texas Health Science Center, Houston, United States; [7]Cinvestav-IPN, Mexico; [8]Department of Biochemistry and Molecular Biology, University of Chicago, Chicago, United States

**\*For correspondence:** francisco. barona@cinvestav.mx

**Present address:** [†]Department of Biological Chemistry, Weizmann Institute of Science, Rehovot, Israel; [‡]Department of Molecular and Cell Biology, University of California, Berkeley, United States; [§]Ciencias de la Computación, Centro de Investigación en Matemáticas, Guanajuato, México

**Competing interests:** The authors declare that no competing interests exist.

**Abstract** The connection between gene loss and the functional adaptation of retained proteins is still poorly understood. We apply phylogenomics and metabolic modeling to detect bacterial species that are evolving by gene loss, with the finding that Actinomycetaceae genomes from human cavities are undergoing sizable reductions, including loss of L-histidine and L-tryptophan biosynthesis. We observe that the dual-substrate phosphoribosyl isomerase A or *priA* gene, at which these pathways converge, appears to coevolve with the occurrence of *trp* and *his* genes. Characterization of a dozen PriA homologs shows that these enzymes adapt from bifunctionality in the largest genomes, to a monofunctional, yet not necessarily specialized, inefficient form in genomes undergoing reduction. These functional changes are accomplished via mutations, which result from relaxation of purifying selection, in residues structurally mapped after sequence and X-ray structural analyses. Our results show how gene loss can drive the evolution of substrate specificity from retained enzymes.

## Introduction

Genome dynamics, or the process by which an organism gains or loses genes, plays a fundamental role in bacterial evolution. Acquisition of new functions due to horizontal gene transfer (HGT) or genetic duplications is broadly documented (*Wiedenbeck and Cohan, 2011*; *Blount et al., 2012*). Gene loss has also been implicated in rapid bacterial adaptation after experimental evolution (*Hottes et al., 2013*), but this process has not yet been confirmed in natural populations. Phylogenomics involves the comparative analysis of the gene content of a set of phylogenetically related genomes to expose new insights into genome evolution and function, and this approach has been classically applied to study how gene gain is associated with functional divergence in bacteria (*Treangen and Rocha, 2011*). Here, we propose that bacterial phylogenomics can be similarly applied to study evolution by gene loss (*Albalat and Cañestro, 2016*), specifically where enzymes

are evolving within bacterial species that are undergoing genome decay (*Adams et al., 2014*; *Price and Wilson, 2014*).

The current bias toward in-depth functional analysis of proteins from genomes that are undergoing gene gain by HGT versus gene loss by decay is likely due to two factors. First, in genomes that are undergoing decay, there is a relaxation in the selection pressure that increases mutation rates in functioning proteins as these proteins begin to contribute less to cell fitness (*Wernegreen and Moran, 1999*; *McCutcheon and Moran, 2011*). As a result, these proteins display higher-than-normal mutation rates, making in vitro analysis of protein function a challenge (*Couñago et al., 2006*). Second, there is only a brief window of opportunity to study the evolution of most proteins during genome decay in bacteria. This is because most proteins are monofunctional, and they are rapidly removed from the bacterial genome once they become dispensable due to gene loss. To overcome this limitation, we propose to use a bifunctional enzyme to study the evolution of substrate specificity after gene loss, as these enzymes may continue to operate when only one of their associated metabolic pathways becomes dispensable. We use genome-scale metabolic models to determine when each pathway is lost as well as when they become non-functional (*Henry et al., 2010*).

The phylum Actinobacteria, Gram-positive organisms with high (G+C)-content, are ubiquitous and show one of the highest levels of bacterial metabolic diversity (*Barka et al., 2016*). This phylum is known to display significant metabolic specialization, and phylogenomics has been previously applied to correlate genome dynamics with metabolic pathway evolution and enzyme specialization (*Noda-García et al., 2013*; *Verduzco-Castro et al., 2016*; *Cruz-Morales et al., 2016*). Moreover, within the deep-rooted family Actinomycetaceae, phylogenetic analyses have suggested the occurrence of genome decay (*Zhao et al., 2014*). Furthermore, we have observed that many actinobacterial species lack a *trpF* gene, while retaining a copy of the potentially bi-functional *priA* gene. The PriA enzyme is capable of operating in the L-histidine biosynthesis pathway as HisA, while also functioning in the L-tryptophan biosynthesis pathway as TrpF (*Noda-García et al., 2013*; *Verduzco-Castro et al., 2016*; *Barona-Gómez and Hodgson, 2003*). As such, we suggest PriA is an ideal candidate to study protein evolution during the process of genome decay.

The product of the *priA* gene, which is a *hisA* homolog, catalyzes two analogous isomerizations of structurally similar substrates: (i) the conversion of *N*-(5'-phosphoribosyl)-anthranilate (PRA) into 1-(*O*-carboxyphenylamino)-1'-deoxyribulose-5'-phosphate (CdRP) (TrpF or PRA isomerase activity); and (ii) the conversion of *N*-[(5-phosphoribosyl) formimino]-5-aminoimidazole-4-carboxamide ribonucleotide (ProFAR) into *N*-[(5-phosphoribulosyl)formimino]-5-aminoimidazole-4-carboxamide ribonucleotide (PRFAR) (HisA or ProFAR isomerase activity) (*Barona-Gómez and Hodgson, 2003*). Moreover, evolution of PriA in response to genome dynamics has lead to the appearance of the Sub-HisA and PriB subfamilies, which have been shown to have different substrate specificities (*Noda-García et al., 2013*; *Verduzco-Castro et al., 2016*) (*Figure 1*). While these enzyme subfamilies were respectively discovered using phylogenetics in the genera *Corynebacterium* and *Streptomyces*, no study has ever centered on the metabolic genes associated with *priA* in the deep-branching organisms belonging to the phylum Actinobacteria, such as those of the family Actinomycetaceae where the transition from HisA into PriA must have taken place (*Noda-García et al., 2015*; *Plach et al., 2016*).

Here we exploit the intrinsic features of PriA to explore the link between the evolution of enzyme function and gene loss within the family Actinomycetaceae, which includes many human oral cavity commensal and pathogenic organisms (*Zhao et al., 2014*; *Yeung, 1999*; *Könönen and Wade, 2015*). We classify this bacterial family into four major evolutionary lineages, including two specific for the genus *Actinomyces*. One of these lineages shows extensive gene loss, including *his* and *trp* biosynthetic genes. After the loss of constrictions imposed by the retention of biosynthetic pathways, we found that evolutionary patterns correlate with the sub-functionalization, yet not necessarily specialization, of PriA into two new subfamilies, which we named SubHisA2 and SubTrpF. We support this classification by comprehensive in vivo and in vitro biochemical characterization of a dozen PriA homologs from *Actinomyces* and closely related taxa. X-ray structural analysis and molecular docking simulations were further used to start investigating the evolution of substrate specificity by gene loss in structural grounds. Our results demonstrate that gene loss can drive functional protein divergence, and provide unprecedented insights into the evolution of enzyme substrate specificity in retained enzymes after gene loss in the bacterial genome.

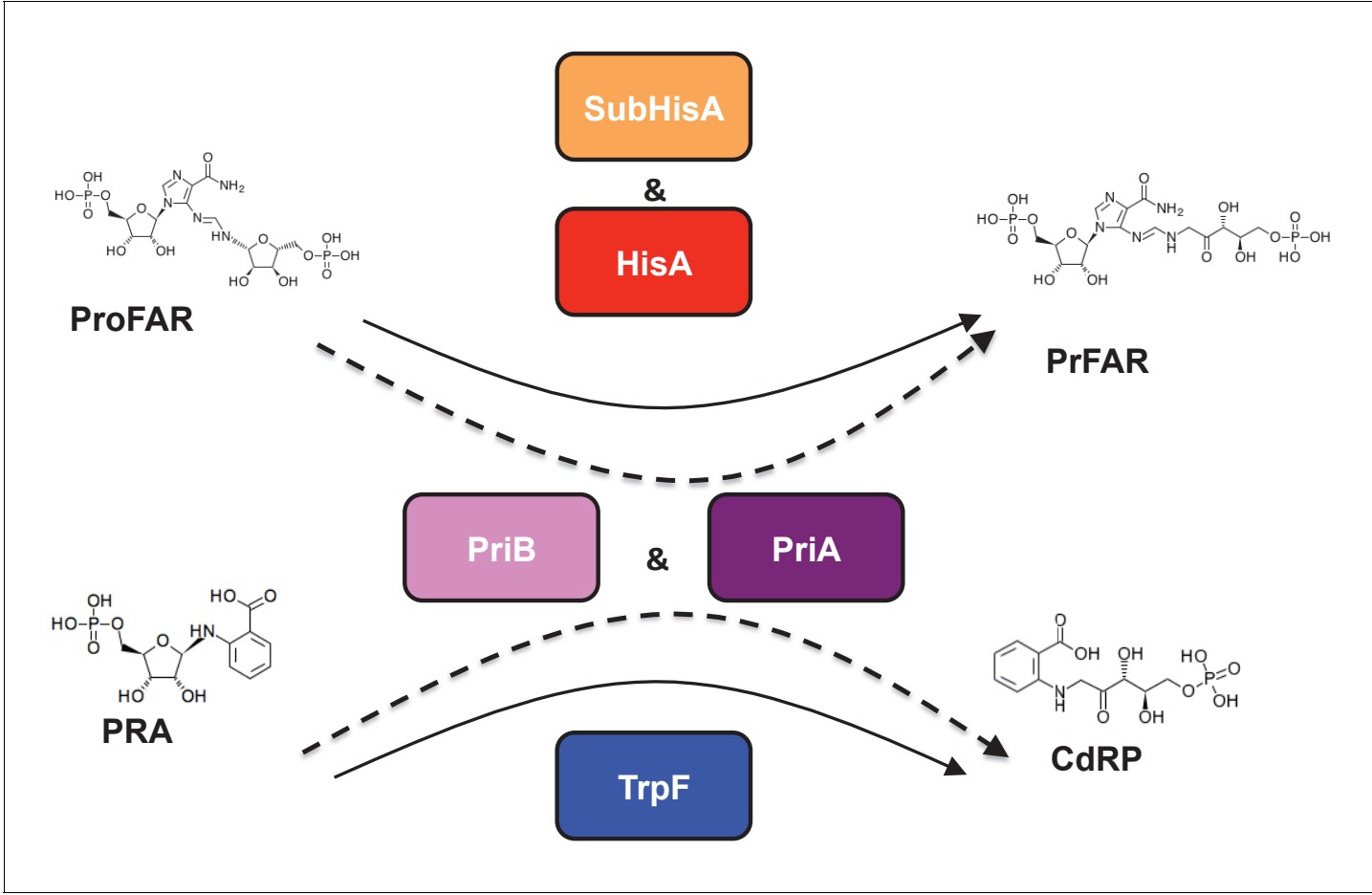

**Figure 1.** $(\beta\alpha)_8$ barrel isomerases at which L-tryptophan and L-histidine biosynthesis converge. Selected L-tryptophan (blue) and L-histidine (red) biosynthetic enzymes are shown. The committed reaction catalyzed by PriA and PriB, or phosporibosyl isomerase A or B in Actinobacteria (dashed arrows), is independently catalyzed by the enzymes TrpF or PRA isomerase, and HisA or ProFAR isomerase (standard arrows) in most bacteria. Furthermore, the SubHisA enzyme, resulting from divergent evolution after an event of HGT and positive selection in certain *Corynebacterium* species, is also shown.

## Results

### Phylogenomic resolution of the *aAActinomycetaceae* family

To find evidence of gene loss in deep-branching organisms of the phylum Actinobacteria, specifically within genera belonging to the order *Actinomycetales*, we selected 133 representative organisms from 18 families with available genome sequences (*Figure 2A*; *Figure 2—source data 1*). We then aimed at resolving their taxonomic relationships using 35 single-copy proteins that are conserved among all 133 genomes analyzed (see Materials and methods and *Figure 2A* – *Figure 2—source data 2*). We concatenated these proteins to reconstruct their phylogeny, and supported the resulting tree by significant Bayesian posterior probabilities. The phylogenetic tree shows several long branches, which correspond to the families Actinomycetaceae, Micrococcaceae, Propionibacteriaceae and Coriobacteriaceae, and to the genus *Tropheryma*. The tree also includes a clade with the family Bifidobacteriaceae as the root of six different sister families, including the Actinomycetaceae (blue branch and grey box in *Figure 2A*).

As expected, all of the organisms contained in the rapidly evolving lineages trended towards smaller genomes and lower (G+C)-content (*Figure 2B*). The Actinomycetaceae genomes were characterized by particularly broad variances in genome size and (G+C)-content, with the variation being most apparent for organisms belonging to the genus *Actinomyces*. Representative organisms of this

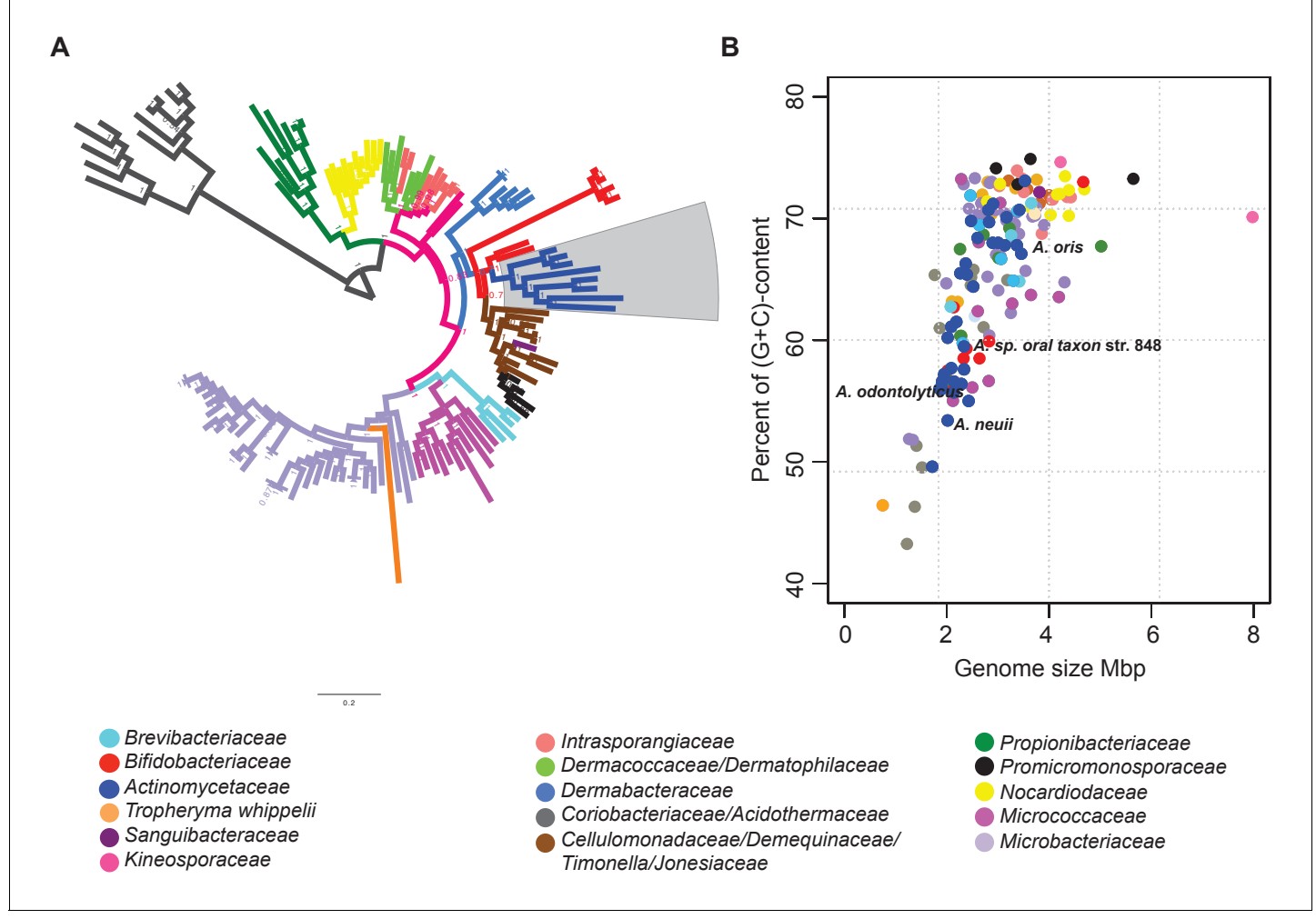

**Figure 2.** Identification of reduced genomes in Actinobacteria. (**A**) Protein-based phylogeny of 133 representative deep-branching Actinobacteria using Bayesian reconstruction. The tree shows a clade with the family Bifidobacteriaceae as the root of the families Dermabacteraceae, Cellulomonadaceae, Demequinaceae, Jonesiaceae, Promicromonosporaceae and Actinomycetaceae, shown in blue and highlighted with a grey box. (**B**) Relationship between genome size and percentage of (G+C) content. The color key used for taxonomic associations is provided at the bottom, and it is the same for both panels.

The following source data is available for figure 2:

**Source data 1.** Actinobacterial genome sequences from early-diverging families used in this study.

**Source data 2.** Conserved orthologs in early-diverging actinobacterial families used for phylogenetic reconstruction.

genus, e.g. *A.* sp. oral taxon 848 str. F0332, *A. oris* MG-1, *A. neuii,* and *A. odontolyticus*, are distributed throughout the Actinomycetaceae clade (blue dots in *Figure 2B*). Given these observations, we selected the genus *Actinomyces* as the ideal target for a deeper analysis of rapid evolution by gene loss.

We then carried out a phylogenomic analysis using the genome sequences of 33 organisms from the family Actinomycetaceae (*Figure 3 - Figure 3—source data 1*), from which 27 are classified as *Actinomyces* (*Zhao et al., 2014*; *Yeung, 1999*), including the model strain *A. oris* MG-1 sequenced in this study. The remaining sequences came from the genera *Actinotignum*, formerly *Actinobaculum* (*Könönen and Wade, 2015*), *Trueperella, Varibaculum* and *Mobiluncus*. As an out-group we used the genera *Bifidobacterium*, which included eight genome sequences. We identified a total of 205 single-copy proteins shared among all these 41 organisms, which were used for constructing a

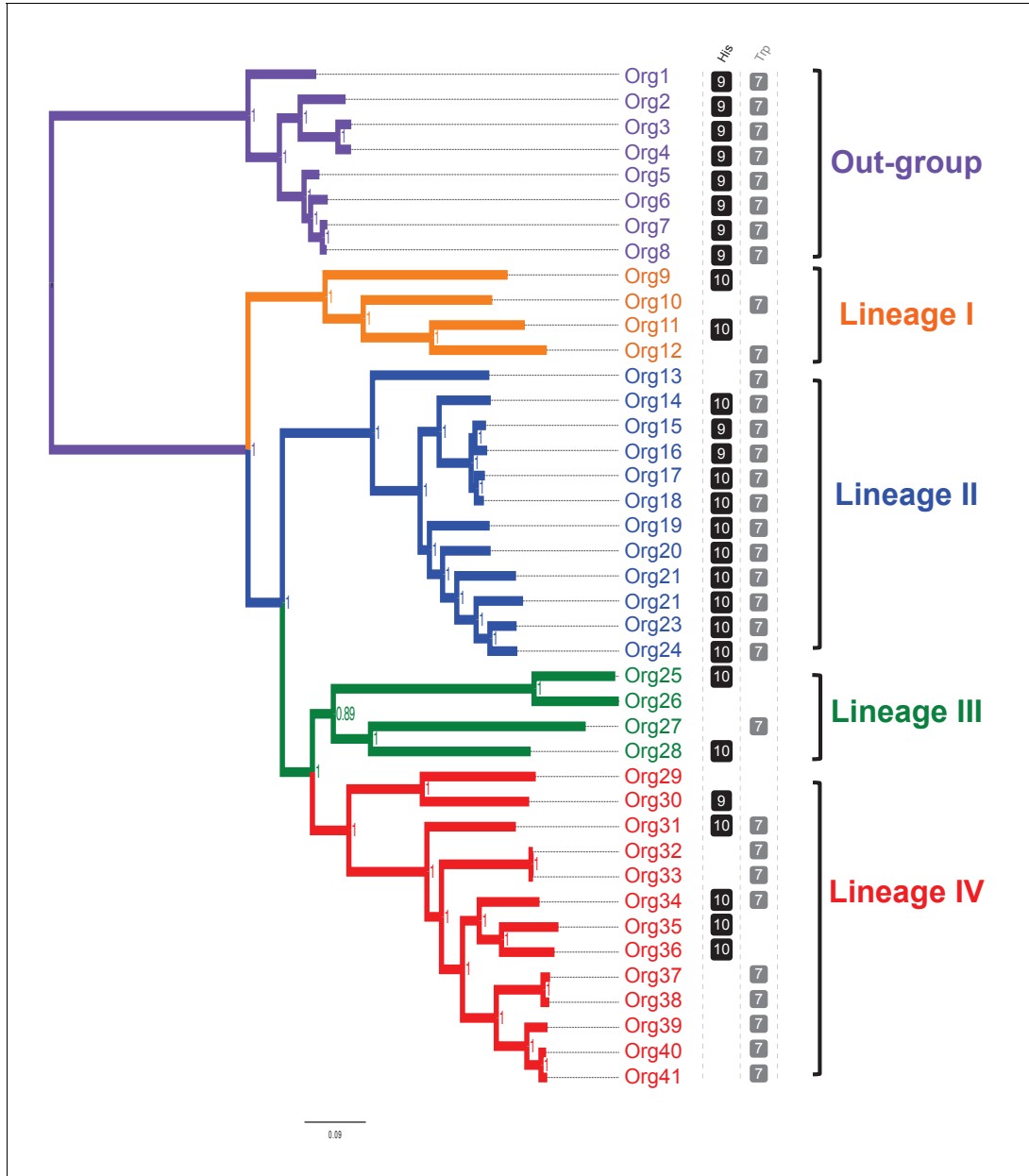

**Figure 3.** Concatenated phylogenetic tree of the family Actinomycetaceae and occurrence of L-histidine and L-tryptophan biosynthetic genes. The tree was constructed using 205 single-copy conserved proteins using Bayesian methods. Only posterior probabilities are shown but significant bootstrap values close to 100 using maximum likelihood were also calculated (*Figure 3—figure supplement 1*). A new classification of the family, into four major groups, is proposed: lineage I (orange); lineage II (blue); lineage III, (green); and lineage IV (red). Based in the species phylogenetic tree of *Figure 2A*, we selected as out-group the genus *Bifidobacterium*. Occurrence of L-histidine (His, black) and L-tryptophan (Trp, grey) biosynthetic genes as revealed by standard genome annotation using RAST is shown next to the tree. Each square represents a complete pathway including all expected genes (10 and 7 for the *his* and *trp* genes respectively) up to 90%. The only missing *his* gene refers to the enzyme histidinol-phosphatase (EC 3.1.3.15), which belongs to a broad enzyme family difficult to annotate.

The following source data and figure supplements are available for figure 3:

**Source data 1.** Genome sequences of the family Actinomycetaceae and the genus *Bifidobacterium* used in this study.

**Source data 2.** Conserved orthologs between the family Actinomycetaceae and the genus *Bifidobacterium* and best fit model used to construct the phylogenetic tree with Mr.Bayes.

**Figure supplement 1.** Concatenated phylogenetic tree of the family Actinomycetaceae using maximum likelihood.

*Figure 3 continued on next page*

*Figure 3 continued*

**Figure supplement 2.** Lineage-specific genomic features of the familiy Actinomycetaceae.

**Figure supplement 2—source data 1.** Statistical analysis of the genomic differences between Lineage II and IV.

concatenated phylogenetic tree by Bayesian (*Figure 3*; *Figure 3—source data 2*), and maximum likelihood approaches (*Figure 3—figure supplement 1*). Based in this analysis, the family Actinomycetaceae separated into four evolutionary lineages contained in three sub-clades: Lineage I, which includes *A.* sp. oral taxon 848 str. F0332 (Org10); lineage II, which includes *A. oris* MG-1 (Org21); and lineages III and IV, which form a monophyletic group and include *A. neuii* (Org27) and *A. odontolyticus* (Org41), respectively. Remarkably, these lineages group depending on their mammalian hosts and human body niches from which they were isolated (*Figure 3* - *Figure 3—source data 1*).

Our phylogenetic analysis also shows that 25 of the 27 *Actinomyces* species analyzed have a paraphyletic origin leading to lineages II and IV. These two lineages can be distinguished not only according to their genome size and (G+C)-content, but also to the number of coding sequences (CDS) and metabolic functions or subsystems (*Figure 3—figure supplement 2*). Specifically, as revealed by genome annotation using RAST (*Aziz et al., 2008*), lineage II, which has the highest (G+C)-content (68.32% on average) and the biggest genome size (3.04 Mbp on average), has the largest number of amino acid biosynthetic pathways (see next section). This observation contrasts with the results obtained for lineage IV, which shows reduced (G+C)-content (60.66% on average) and genome size (2.19 Mbp on average), as well as less amino acid biosynthetic pathways. Indeed, the genomic differences between lineages II and IV were found to be statistically significant, including the presence or absence of the *his* and *trp* biosynthetic genes (*Figure 3*; *Figure 3—figure supplement 2—source data 1*). We explore this observation in more detail by constructing metabolic models for all of the analyzed genomes in the following section.

## Metabolic evolution of the Actinomycetaceae family

In order to reduce the risk of overreaching conclusions based only in homology sequence searches, we constructed genome-scale metabolic models of all 33 organisms comprising this family, plus the nine outgroup *Bifidobacterium* species (see Materials and methods). Next, flux balance analysis was applied to predict the minimal nutrients required to support growth for each genome. Finally, after automated curation of the metabolic reconstructions (*Satish Kumar et al., 2007*), which includes not only homologous but also analogous enzymes, we classified each reaction in each model as: (i) essential for growth on predicted minimal media; (ii) functional but not essential; and (iii) nonfunctional. All model results, which represent the highest quality functional annotation available for metabolism, are provided as source data of *Figure 4*: model overview (*Figure 4—source data 1*), reaction content and classifications (*Figure 4—source data 2*) and predicted minimal media (*Figure 4—source data 3*).

The lineage II models were generally the largest with an average of 1019 gene-associated reactions, which is to be expected since the lineage II genomes are also the largest. These models also had the fewest predicted essential nutrients with an average of 19 nutrients required. This result indicates that most biosynthetic pathways for essential biomass precursors are complete in the lineage II models. The lineage I and IV models were substantially smaller with an average of 850 and 843 gene-associated reactions, respectively. Although similar in size, the lineage I models had more required nutrients (25 on average) compared with the lineage IV models (22 on average). Finally, the lineage III models were the smallest of all, with an average of 817 gene-associated reactions. Surprisingly, these models still had fewer required nutrients than the lineage I models (23 on average). These results provide a meaningful biochemical context in which biosynthetic enzymes are evolving.

To study the metabolic diversity of each lineage in more detail, we performed a comparative analysis of the gene-associated reactions of our models (*Figure 4A–D*). Given the large metabolic and genetic diversity, we used less stringent parameters than those used for our core genome analysis sustaining our phylogenomics of previous section (see Materials and methods). This comparative

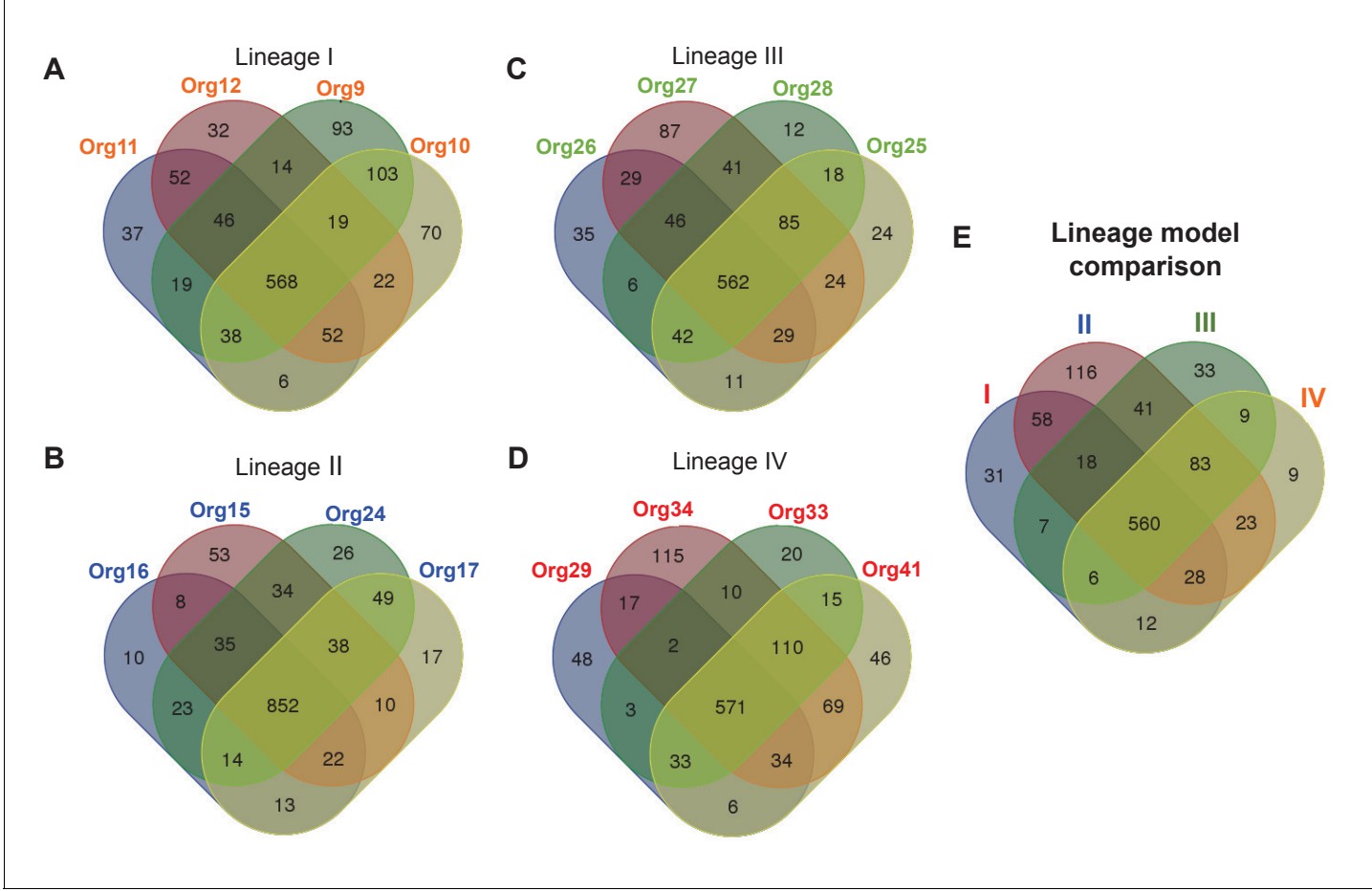

**Figure 4.** Metabolic diversity amongst the genomic lineages of the genus *Actinomyces*. The Venn diagrams show the overlap in gene-associated reactions included in models of genomes of lineage I (A), lineage II (B), lineage III (C), and lineage IV (D). The diagrams for lineages I and III show the overlap of all models in these lineages, while the lineage II and IV diagrams show the overlap of a subset of models sampled based on their metabolic diversity. Overlap in gene-associated reaction content for each of the core lineage models (E), which are comprised of conserved reactions present in at least 75% of the models in each lineage, is also shown.

The following source data and figure supplement are available for figure 4:

**Source data 1.** Model overview.
**Source data 2.** Model reactions.
**Source data 3.** Predicted minimal media.
**Figure supplement 1.** Phylogenetic projection of amino acid biosynthetic pathways throughout the family Actinomycetaceae as confirmed after genome-scale metabolic modeling.

analysis revealed that the lineage II genomes were the least diverse, with a very large fraction of reactions present in all models, including those for amino acid biosynthesis (*Figure 4—figure supplement 1*). All other lineages were more diverse. Interestingly, a comparative analysis of our models found that all models across all lineages share a common conserved core of 695 reactions. When we similarly compute a conserved core for each individual lineage (*Figure 4E*), we find that the 89% of reactions in the conserved core for each lineage are contained in the conserved core across all lineages.

From these modeling results, we clearly see that lineages I, III and IV are all undergoing the process of gene loss, resulting in a reduction towards a common set of core metabolic pathways. This

explains the rapid development of diversity within each lineage, as well as the variability in minimal required nutrients predicted by our models. We can also apply our models to study the gene loss process from a mechanistic perspective by looking for patterns in the presence and absence of genes and reactions for two specific pathways of interest: L-tryptophan and L-histidine biosynthesis. Our models predicted genomes in lineages I, III, and IV (but none from lineage II) that required these amino acids as a supplemental nutrient, indicating the loss of these biosynthetic pathways in these organisms. We also observed that the presence of the *priA* gene, which takes part in both L-tryptophan and L-histidine biosynthesis, closely tracked with the presence of these pathways in these genomes (*Figure 5A*). This observation suggests that gene losses could have an effect on the evolution of the retained PriA enzymes.

## Molecular evolution of PriA within the family Actinomycetaceae

To bring down these observations at the enzyme level, we carried out comparable phylogenetic analyses of PriA (*Figure 5*), and we measured the evolutionary rate of its gene by estimating the $d_N/d_S$ ratio ($\omega$ value) for each resulting clade (*Table 1*). The PriA phylogeny was complemented with an analysis of the occurrence of the *his* and *trp* biosynthetic genes, including *priA* for both pathways (*Figure 5*; *Figure 5—source data 1*). Excluding the out-group, our phylogenetic reconstructions show that PriAs from different lineages are grouped in three sub-clades, highlighted in purple, orange, and yellow boxes in *Figure 5A*, which have distinguishable selection pressures operating upon them. This analysis also shows that PriA coevolves with the presence or absence of the *his* and *trp* biosynthetic genes (*Figure 5B*).

The purple box denotes a paraphyletic clade that includes PriAs from lineage II, as well as the PriAs from the genus *Bifidobacterium* used as an out-group. The $d_N/d_S$ value of this lineage, which retains the entire set of *his* and *trp* genes, is 0.0636, consistent with purifying selection. The orange and yellow boxes denote polyphyletic groups that include PriAs from lineages I, III, and IV. Interestingly, the included taxa within these lineages lost their extant *his* or *trp* genes differentially (*Figure 4A*), and their $d_N/d_S$ values are 0.0901 and 0.1459, respectively, which is suggestive of relaxation of purifying selection. Moreover, the higher $d_N/d_S$ values in the clade shown in yellow seem to be due to accumulation of nonsynonymous substitutions, in other words, higher values of $d_N$ that may relate to changes in enzyme specificity (*Table 1*).

Thus, on the basis of these evolutionary observations, we proposed three functional and testable hypotheses related to the emergence of novel PriA enzyme subfamilies in the bacterial family Actinomycetaceae (*Figure 6A*). In H1 (purple box) we assume that PriA homologs are conserved as enzymes with dual-substrate specificity, capable of converting both PRA and ProFAR substrates. In H2 (orange box) and H3 (yellow box) the PriA homologs are expected to be monofunctional isomerases, yet not necessarily specialized enzymes, capable of converting ProFAR or PRA as substrates, respectively. Moreover, given that relaxation of purifying selection is associated with the latter two hypothetical scenarios, H2 and H3, our model predicts monofunctional, yet not necessarily specialized enzymes capable of supporting growth. Representative enzymes of each hypothesis were selected for further biochemical characterization, as described.

## Biochemical confirmation of the evolution of PriA by gene loss

Before evaluating the functional implications of our evolutionary hypotheses from previous section, we confirmed that the *priA* gene is functional in *Actinomyces*. For this purpose, we used allelic exchange to delete the *priA* gene from the chromosome of *A. oris* MG-1 (Org21) (*Wu and Ton-That, 2010*; *Delisle et al., 1978*), a model strain that belongs to lineage II and whose genome was

**Table 1.** Selective pressures in PriA homologs from H1, H2 and H3 hypotheses.

| Hypothesis | $d_N/d_S$ | $d_N$ | $d_S$ |
| --- | --- | --- | --- |
| H1 | 0.0636 | 0.3151 | 4.9559 |
| H2 | 0.0901 | 1.8687 | 20.736 |
| H3 | 0.1459 | 1.8703 | 12.8227 |

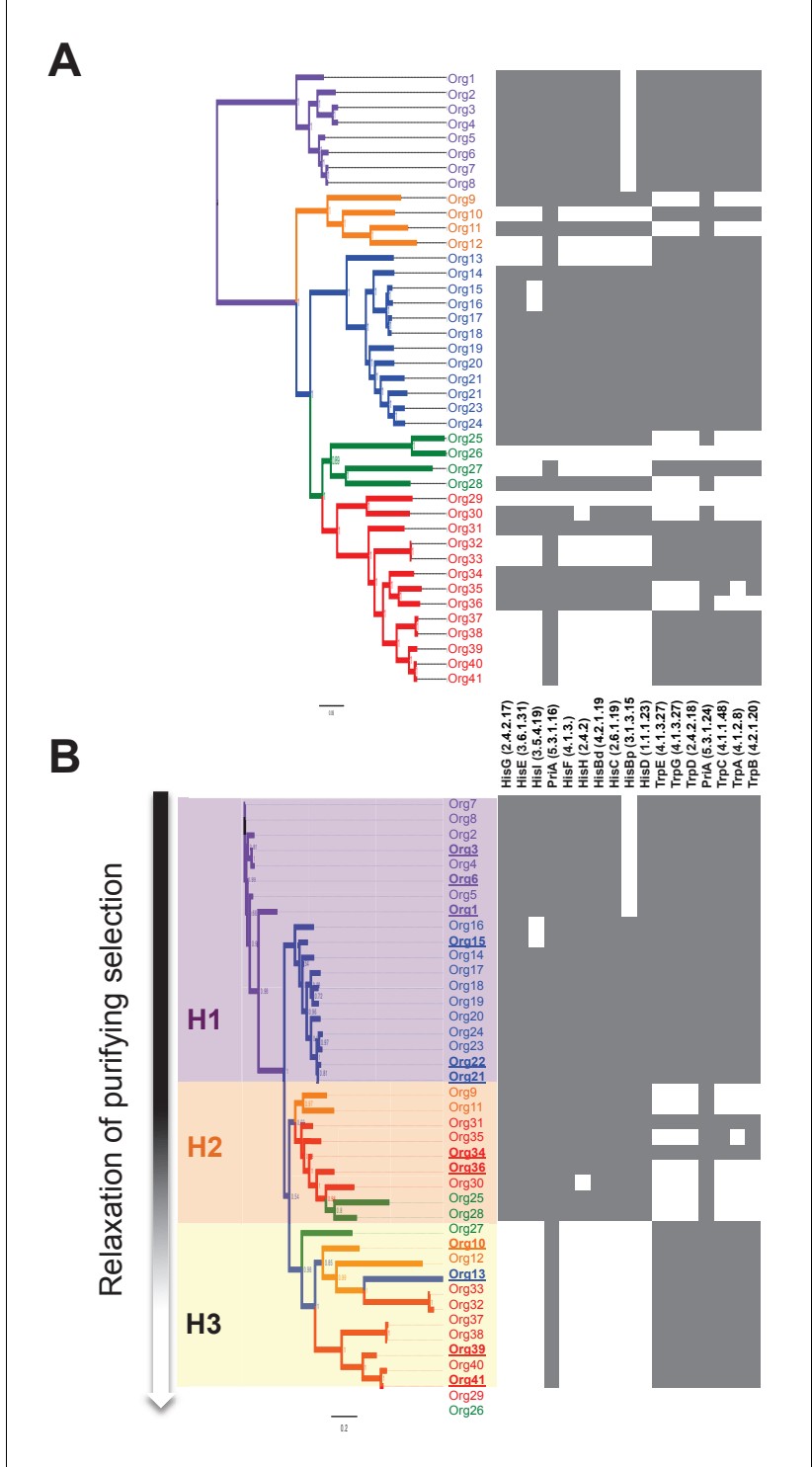

**Figure 5.** Phylogenetic reconstruction of PriA and coevolution with L-histidine and L-tryptophan biosynthesis. (**A**) Analysis of the occurrence of *his* and *trp* biosynthetic genes (*priA* is included in both pathways), marked as absent (white) or present (gray), using the phylogenomics species tree of *Figure 3* as a map (same color code). The missing *his* gene, when almost the entire pathway is present, refers to the enzyme histidinol-phosphatase (EC 3.1.3.15), which belongs to a broad enzyme family difficult to annotate. (**B**) Same gene occurrence analysis using the PriA phylogenetic tree as a map. Three evolutionary scenarios where PriA is coevolving with the occurrence of *his* and *trp* genes, and in agreement with the intensity of purifying selection (*Table 1*, gradient shown in the left-

*Figure 5 continued on next page*

*Figure 5 continued*
hand side of the panel), are marked as H1 (purple), H2 (orange), and H3 (yellow). The same color code as in
*Figure 3* is used, and the selected enzymes that were biochemically characterized are underlined.
The following source data is available for figure 5:

**Source data 1.** Occurrence of L-Histidine and L-Tryptophan biosynthetic enzymes throughout the family Actinomy-
cetaceae and the genus *Bifidobacterium*.

sequenced as part of this study. Mutation of *priA* in this organism was confirmed by sequencing the entire genome of the resulting *ΔpriA* mutant strain (*Supplementary file 1*). Determination of the growth requirements of this strain, termed *ΔpriA_Org21*, showed that *priA* mutation leads to L-tryptophan auxotrophy, demonstrating the physiological relevance of PriA in this organism. Unexpectedly, however, the *ΔpriA* mutant remains prototrophic for L-histidine, which could not be explained on the basis of current data. Thus, it is tempting to speculate that this phenotype may find an explanation in the previously reported association between enzyme promiscuity and genome decay (*Adams et al., 2014*; *Price and Wilson, 2014*).

To biochemically evaluate the functional implications of our evolutionary hypotheses (*Figure 6A*), we characterized nine selected PriAs, both in vivo, by complementation assays using *trpF* and *hisA Escherichia coli* mutants; and in vitro, by estimation of their Michaelis Menten steady-state enzyme kinetic parameters, as we have previously done (*Noda-García et al., 2013*; *Verduzco-Castro et al., 2016*; *Noda-García et al., 2010*). The results of these experiments are included in *Table 2*. First, in vivo complementation assays using appropriate *priA* constructs, showed that PriA homologs from Org15, Org21, and Org22 (H1) were able to rescue growth of both HisA and TrpF deficient strains. Second, *priA* homologs from Org34 and Org36 (H2) complemented the HisA activity and, to a lesser extent, the TrpF activity. Third, those *priA* homologs from Org10, Org13, Org39 and Org41 (H3) were able to complement the TrpF activity but not the HisA activity.

The *priA* homologs were then heterologously expressed and purified to homogeneity in *E. coli* (see Materials and methods). Only five enzymes out of nine were found to be soluble and could be purified as needed, which agrees with the high mutation rate encountered in the previous section. Fortunately, we obtained Michaelis Menten enzyme kinetics parameters for representative enzymes of all three evolutionary hypotheses, namely, three enzymes belonging to H1 and one enzyme each for H2 and H3, with the following results (*Figure 6B* and *Table 2*). First, enzymes from Org15, Org21, and Org22 (H1) showed dual-substrate specificity but also poor catalytic efficiencies, namely, $k_{cat}/K_M^{ProFAR}$ from 0.01 to 0.1 $\mu M^{-1}s^{-1}$ and $k_{cat}/K_M^{PRA}$ around 0.01 $\mu M^{-1}s^{-1}$. Second, only ProFAR isomerase activity could be detected in vitro using pure enzyme from Org36 (H2), with a catalytic efficiency of $k_{cat}/K_M^{ProFAR}$ of 0.002 $\mu M^{-1}s^{-1}$, but not PRA isomerase activity, as suggested by our highly sensitive in vivo complementation assay. Third, PRA isomerase activity as the sole activity present in H3 was confirmed in the enzyme purified from Org42, with a $k_{cat}/K_M^{PRA}$ of 0.02 $\mu M^{-1}s^{-1}$.

The obtained enzyme kinetics parameters suggest that mutations that accumulate during relaxation of purifying selection, which make these enzymes difficult to work with, affect the turnover ($k_{cat}$). In the case of the H1 enzymes, the poor turnovers are compensated for by relatively high substrate affinities ($K_M$), mainly for ProFAR. However, this does not seem to be the case for the enzymes belonging to H2 and H3, which have poor $K_M$ parameters not only for the substrate of the missing activity but also for the substrates they are active against, ProFAR and PRA, respectively. Therefore, PriA homologs from *Actinomyces* have poor catalytic efficiencies when compared with *bona fide* PriAs from its closely related genus *Bifidobacterium* (*Table 2*). This suggests that enzyme evolution from bifunctionality to monofunctionality under relaxation of purifying selection does not necessarily express itself in the same way as recorded during purifying or positive selection, where specialization and enzyme proficiency come together.

The case of the in vivo PRA isomerase activity detected for the enzyme from Org36, which could not be confirmed in vitro, may be related to the different resolutions of our enzyme assays. For instance, the detection limits for the PRA and ProFAR isomerase assay used in the present study are 0.0001 $\mu M^{-1}s^{-1}$ and 0.001 $\mu M^{-1}s^{-1}$, respectively (*Noda-García et al., 2013*; *Verduzco-Castro et al., 2016*; *Noda-García et al., 2010*). However, despite the poor catalytic efficiency of all

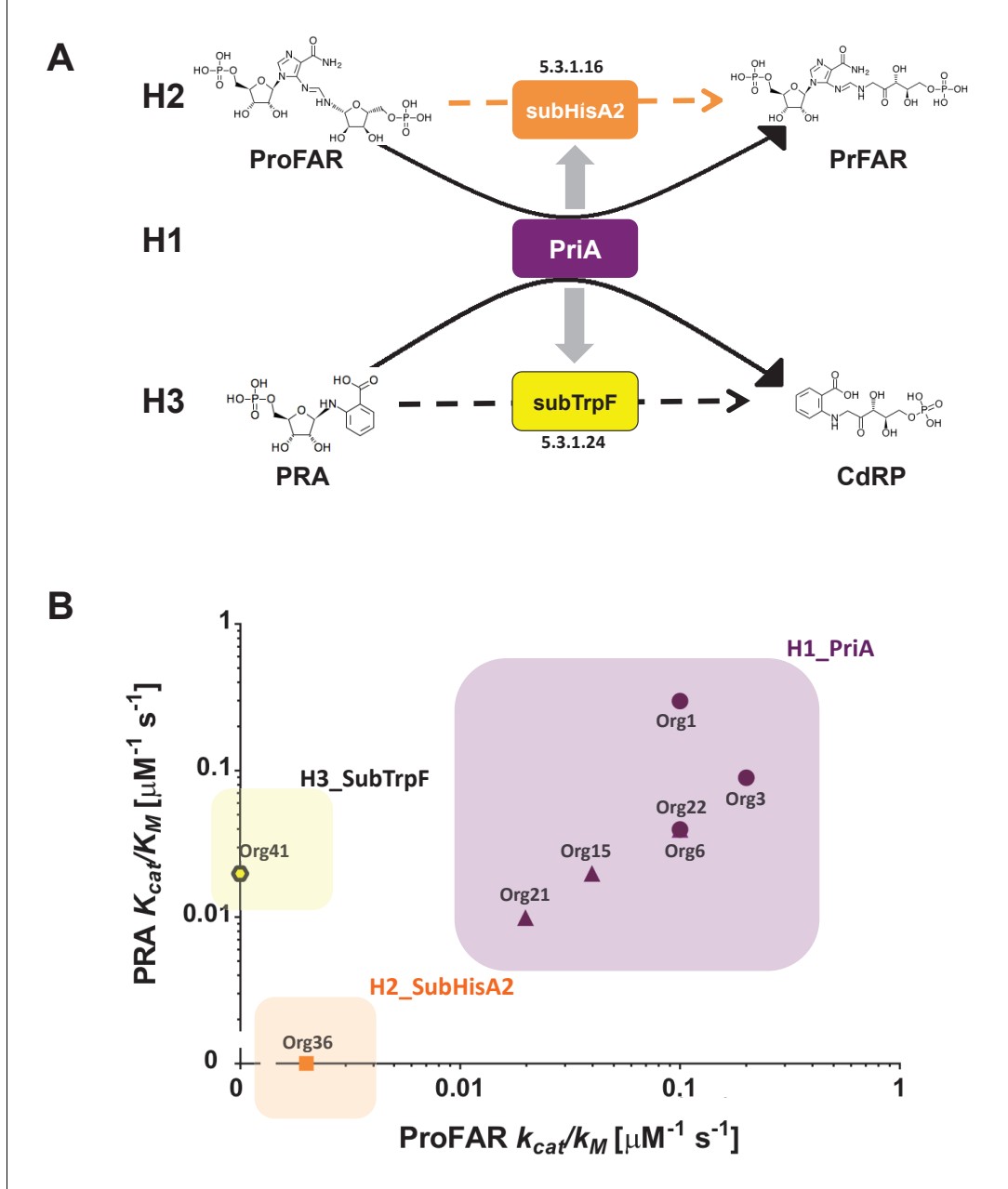

**Figure 6.** Evolutionary hypotheses and steady-state enzyme kinetics of PriA homologs. (**A**) Evolutionary hypothesis (H1, H2 and H3) with functional implications leading to PriA enzyme subfamilies, expressed as biochemical conversions, as obtained from *Figure 5*. (**B**) Comparison of the catalytic efficiencies (*kcat/K$_M$*) of selected enzymes from different scenarios, including the three postulated evolutionary hypotheses. Values for ProFAR (x axis) and PRA (y axis) isomerase activities, expressed as log10, are compared. Data from PriAs of *Bifidobacterium* (purple circle), PriA from H1 (purple triangle), SubHisA2 from H2 (orange), and SubTrpF from H3 (yellow pentagon) is included.

*Actinomyces* enzymes investigated, these detection limits guarantee that our enzyme parameters are in agreement between them and with our hypotheses. Based on these results the family related to H1, which has both activities, is referred to as PriA, whereas the latter two enzyme subfamilies, related to H2 and H3, were renamed as SubHisA2 and SubTrpF, respectively. These names, together with the name of the organism from which the enzymes were obtained, are used in *Table 2* and in the following sections.

## Structural insights into the evolution SubHisA2 and SubTrpF

To potentially identify mutations in active-site residues that may affect $k_{cat}$ and $K_M$ parameters, we attempted to elucidate the structure of the five PriA homologs that we were able to in vitro characterize. However, we were only able to crystallize and solve the structure of PriA_Org15 (H1) at atomic resolution of 1.05 Å (PDB: 4 × 2R, *Figure 7*; *Figure 7—source data 1*). To compare this structure with SubHisA2_Org36 (H2) and SubTrpF_Org41 (H3), we opted for the construction of structural homology models. Since the ability of PriA to accept both ProFAR and PRA as substrates requires productive conformations, we also explored these interactions using molecular docking. This was complemented with detailed structure-based multiple sequence alignments taking into account all available PriA functional and structural data (*Figure 7B*). This combined approach allowed us to identify mutations that may be driving the evolution of PriA into SubHisA2 and SubTrpF enzyme subfamilies.

Changes of conserved residues from PriA (H1) into SubHisA2 (H2) enzymes include Ile47Leu and Ser79Thr. Previous independent mutation of these two residues, even into similar amino acids, in SubHisA from *Corynebacterium* abolished the PRA isomerase activity of this monofunctional enzyme (*Noda-García et al., 2013*). Analogously, the SubHisA2_Org36 has a change of Ser79 into Thr79 (*Figure 7B*). In this mutation, the methyl group of the threonine residue may affect the contact between PRA and the hydroxyl group common to these residues (*Figure 7A*), thus abolishing PRA isomerase activity. This effect agrees with the estimated binding affinities for ProFAR (−9.5 kcal/mol) and PRA (−9.2 kcal/mol) obtained after molecular docking (*Supplementary file 2*). The energy-minimized docking model of the productively bound PRA, in agreement with the kinetic parameters from the preceding section, indicates that the catalytic residue Asp11 does not interact with the 2′-hydroxyl group from the substrate. A precedent for this contact is found in previous X-ray structural and mutagenesis analysis of *bona fide* PriA enzymes (*Noda-García et al., 2010*; *Due et al., 2011*).

**Table 2.** Biochemical characterization of PriA, SubHisA2 and SubTrpF homologs.

| | In vivo activity | | In vitro activity * | | | | | | |
| | | | ProFAR isomerase (HisA) | | | PRA isomerase (TrpF) | | |
| Enzymes | HisA | TrpF | $K_M$ (μM) | $k_{cat}$ (s$^{-1}$) | $k_{cat}/K_M$ (s$^{-1}$μM$^{-1}$) | $K_M$ (μM) | $k_{cat}$ (s$^{-1}$) | $k_{cat}/K_M$ (s$^{-1}$μM$^{-1}$) |
|---|---|---|---|---|---|---|---|---|
| PriA_Org3_*B. longum* | + | + | 2.7 ± 0.5 | 0.4 ± 0.1 | 0.1 | 6.1 ± 0.1 | 2.1 ± 0.5 | 0.3 |
| PriA_Org1_*B. gallicum* | + | + | 1.7 ± 0.3 | 0.3 ± 0.1 | 0.2 | 40 ± 9 | 3.5 ± 0.1 | 0.09 |
| PriA_Org6_*B. adolescentis* | + | + | 17 ± 4.3 | 2.3 ± 0.01 | 0.1 | 21 ± 5 | 0.9 ± 0.2 | 0.04 |
| PriA_Org15_*A. urogenitalis* | + | + | 4.0 ± 0.9 | 0.2 ± 0.03 | 0.04 | 23 ± 6.5 | 0.5 ± 0.05 | 0.02 |
| PriA_Org22_*A. sp. oral* taxon 171 | + | + | 3 ± 0.3 | 0.3 ± 0.09 | 0.1 | 8 ± 2 | 0.4 ± 0.1 | 0.04 |
| PriA_Org21_*A. oris* MG-1 | + | + | 10 ± 2 | 0.2 ± 0.09 | 0.02 | 30 ± 7 | 0.3 ± 0.03 | 0.01 |
| SubHisA2_Org34_*A. vaccimaxillae* | + | + | | | | | | |
| SubHisA2_Org36_*A. cardiffensis* | + | + | 56 ± 17 | 0.14 ± 0.05 | 0.002 | n.d. | n.d. | n.d. |
| SubTrpF_Org10_*A. sp. oral* taxon 848 | − | + | n.d. | n.d. | n.d. | n.d. | n.d. | 0.0001 |
| SubTrpF_Org13_*A. graevenitzii* | − | + | | | | | | |
| SubTrpF_Org39_*A. sp. oral* taxon 180 | − | + | | | | | | |
| SubTrpF_Org41_*A. odontolyticus* | − | + | n.d. | n.d. | n.d. | 8.5 ± 0.9 | 0.15 ± 0.06 | 0.02 |

* Each data point comes from at least three independent determinations using freshly purified enzyme. n.d., activity not detected, even using active-site saturation conditions. Empty entries reflect our inability to properly express and/or solubilize these proteins. The detection limits for the PRA and ProFAR isomerase assay used in the present study are 0.0001 μM$^{-1}$s$^{-1}$ and 0.001 μM$^{-1}$s$^{-1}$, respectively (*Noda-García et al., 2013*; *Verduzco-Castro et al., 2016*; *Noda-García et al., 2010*).

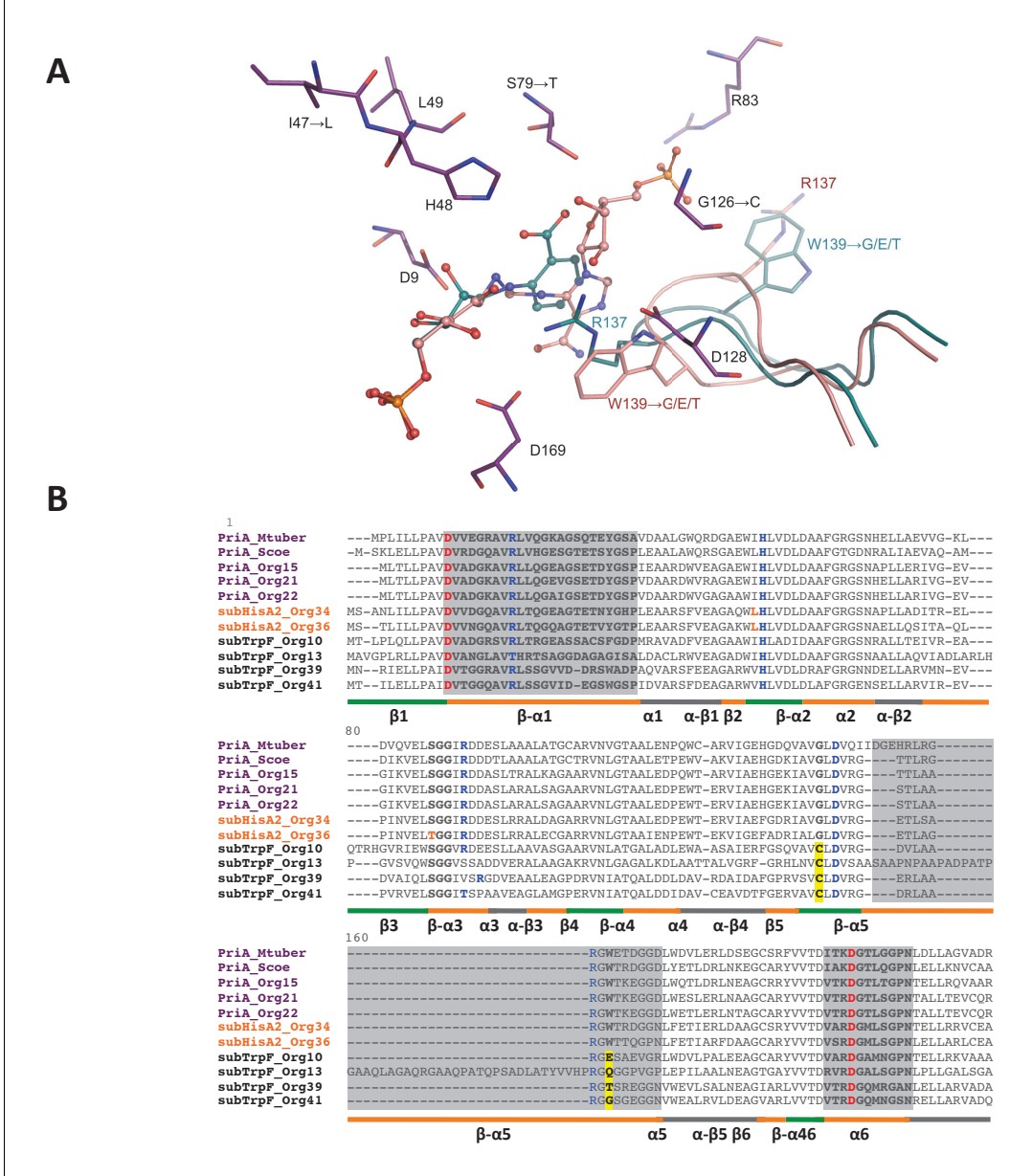

**Figure 7.** PriA from Org15_*A. urogenitalis* active site and sequence alignment of PriA sub-families. (**A**) The structure of PriA from *A. urogenitalis* (purple, PDB: 4X2R) superimposed with PriA from *M. tuberculosis* in a complex with rCdRP (cyan, PDB: 2Y85) and PrFAR (pink, PDB: 2Y88) is used to illustrate the position of the respective substrates. The catalytic residues and those critical for divergence into SubHisA2 or SubTrpF are shown. Since the loop contributing Trp139 and Arg137 is mostly disordered, and Arg137 itself does not adopt substrate binding-relevant position in the structure from *A. urogenitalis*, only the equivalent elements from the *M. tuberculosis* homolog are shown. (**B**) Multiple sequence alignment of PriA (purple), SubHisA2 (red) and SubTrpF (bold) sequences. Catalytic residues, Asp9 and Asp169, are marked in red. PRA and ProFAR binding residues are shown in blue. SubHisA2 and SubTrpF loss-of-function residues are framed. The secondary structure is shown below the sequences. Loops are shown in orange, α helixes are shown in gray and β sheets are shown in green. Sequence corresponding to loops 1, 5, and six is highlighted in gray. List of Tables provided as Source Data.

The following source data is available for figure 7:

**Source data 1.** X-ray crystalographic data processing and refinement statistics for PriA_Org15.

Comparison of PriA (H1) with SubTrpF (H3) revealed the mutations Gly126Cys and Trp139Gly. In PriA, Gly126 faces the active site near the catalytic residue Asp128. The introduction of the Cys side-chain in SubTrpF could influence the positioning of Arg137 with respect to Asp128, obstructing the accommodation of ProFAR, as this region interacts with a large phosphosugar moiety that is absent from PRA (*Figure 7B*). Furthermore, Trp139, which is catalytically important for conversion of Pro-FAR by PriA, is mutated into several different amino acid residues in SubTrpFs. Trp139 is important for the correct positioning of the catalytic residues present in loop 5, and for substrate binding through stacking interactions (*Verduzco-Castro et al., 2016*; *Due et al., 2011*). Indeed, the indole group of Trp139 in PriAs can form a hydrogen bond with Asp128, stabilizing the knot-like conformation observed during ProFAR binding. Thus, mutation of this residue in SubTrpF is in agreement with the loss of ProFAR isomerase activity. Arg83 is also interesting as it is differentially missing from SubTrpF, and/or the fragment preceding it contains a two-residue insertion (*Figure 7B*). Arg83 inter-acts with the second phosphate group of ProFAR, allowing its correct position in the substrate-bind-ing pocket of PriA. Overall, these modifications in key residues disfavor the ProFAR binding affinities, a result that is in agreement with the enzyme kinetic parameters and the estimated binding affinities for ProFAR (−9.5 kcal/mol) and PRA (−9.7 kcal/mol) obtained after molecular docking (*Supplementary file 2*).

Although further research is needed to confirm the exact mutations and their roles leading to SubTrpF and SubHisA2 sub-families, our results provide a promising first step towards deciphering at the atomic level how relaxation of purifying selection influences the evolution of substrate specific-ity. At this point in time, when PriA, SubTrpF and SubHisA2 sequences and structures are still scarce, the effects of genetic drift, i.e. mutations related to taxonomic distance rather than functional diver-gence (as previously shown for the evolution of PriB from PriA [*Verduzco-Castro et al., 2016*]) can-not be ruled out. An extra factor potentially hampering sequence and structural analysis is the higher-than-normal mutation rates of these protein sub-families, which translates into lack of sequence conservation and disordered regions in X-ray crystal structures. Our structural data, includ-ing the estimates for molecular binding affinities, can therefore only be used to support other bio-chemical and evolutionary evidence.

## Discussion

Our study highlights the use of phylogenomics and metabolic models to identify and investigate gene loss in bacteria. Our results indicated that the distinctive reactions retained in each *Actinomy-ces* genome reflect the preservation of some full biosynthetic pathways over others, conferring a capacity to grow on different sets of environmental nutrients. This result in turn implies an exposure of these genomes to a diverse range of environmental conditions and selection pressures, while the phylogenetic proximity of these functionally diverse genomes speaks to a strong capacity for rapid adaptation to the diverse conditions present in the human body. The process of gene loss, associ-ated with relaxation of purifying selection, is the key driver of this adaptation strategy. Thus, meta-bolic diversity in complex systems as the human microbiome might be characterized after reconstruction of evolutionary trajectories, which may reflect different bacterial functions and eco-logical sub-niches. The pattern of reaction conservation seen in our metabolic modeling analysis exemplifies a likely signature for gene loss, which could be used to identify these phenomena among other genome families. Remarkably, in this context, enzyme specialization does not necessarily means catalytic proficiency.

Our study of this gene loss process exposed evolutionary patterns of PriA in L-tryptophan and L-histidine biosynthesis pathways, with the potential to unveil the underpinning mechanisms driving the evolution of substrate specificity of retained enzymes. Because multifunctional enzymes may have more than one constraint operating on them, tracking functional evolution promptly after selec-tion is relaxed during genome decay might be done more readily than with monofunctional enzymes. As shown here, only partial selection may be released in the retained bifunctional enzyme PriA. Indeed, the predicted metabolic phenotypes unveiled by flux balance analysis did correlate better with the evolutionary patterns revealed by metabolic gene occurrence and PriA phylogenetic reconstructions than they did with the natural history told by the species tree. To confirm this sort of evolutionary behavior, other instances of well-known multifunctional proteins, such as moonlighting proteins, may be investigated.

The occurrence of SubHisA2 in *Actinomyces*, together with the appearance of SubHisA in *Coryne-bacterium*, demonstrates that subfunctionalization of PriA leading into HisA-like enzymes has occurred at least twice. Such phenotypic plasticity is a reflection of the intrinsic enzymatic proficiency of PriA upon two related but topologically dissimilar substrates; but, more interestingly, the evolutionary histories behind these independent subfunctionalization events responded to somehow contrasting evolutionary mechanisms. Whereas SubHisA is the result of positive selection after the acquisition of an entire *trp* operon by HGT (*Noda-García et al., 2013*), SubHisA2 responded to the loss of *trp* genes, and it evolved under relaxation of purifying selection. Consequently, SubHisA has drastic mutations in its catalytic active site, which have been shown to be responsible for its inability to catalyze PRA, whilst SubHisA2 shows some residual PRA isomerase activity, congruent with the observation that its active-site architecture is almost completely conserved.

The subfunctionalization of PriA into SubTrpF, in contrast, has been documented only here. This functional shift had to involve 'non-proficient' enzyme specialization, since the ancestral activity of PriA is ProFAR isomerase (*Plach et al., 2016*). Thus, the appearance of SubTrpF with substitutions in its catalytic active-site could be discussed based on previous knowledge about PriA. These mutations actually resulted in the elimination of the ancestral ProFAR activity, which is remarkable because the driving force behind this process relates to the relaxation of purifying selection. In agreement, a recent study of PriA sequences obtained from a diverse metagenome, complemented by some of the SubTrpF sequences studied here, classified this enzyme subfamily at the transition from HisA into PriA (*Noda-García et al., 2015*). Since *Actinomyces* undergoes interspecies recombination with protein functional implications (*Do et al., 2008*), such a mechanism may provide a means to explain the sequence heterogeneity found in these *Actinomyces* PriA homologs.

Our study, therefore, provides experimental evidence that gene loss can drive functional protein divergence. It also shows that, despite the fact that the retained enzymes possess low catalytic activities, they contribute to the maintenance of metabolism, and therefore, to fitness. Taken together, our evolutionary observations backed with metabolic modeling, biochemical and structural data, suggest multiple selection types associated with ecological micro-niches, e.g. environmental cues provided by the human body. Thus, enzyme subfamilies are the result of processes involving different selection types upon proteins with more than one function. Although further examples showing metabolic-driven evolutionary histories need to be identified, our study provides a strategy for the in-depth use of genome sequences for protein and bacterial evolutionary studies to understand enzyme function.

# Materials and methods

## Phylogenomic and evolutionary analysis

The genomes of the genus *Bifidobacterium* and the family *Actinomyceatceae* were obtained from NCBI (NCBI accession numbers are provided as *Figure 3—source data 1*). The genomes were annotated by using RAST (*Aziz et al., 2008*). We identified core orthologous genes using BBHs (*Tatusov et al., 1997*) with a defined e-value of 0.001. The sequences were aligned with MUSCLE 3.8.31 (*Edgar, 2004*) and edited with GBLOCKS (*Castresana, 2000*). We concatenated all the orthologous groups for phylogenomic analysis. The phylogenetic analyses were carried out using MrBayes v.3.2.1 (32) and maximum likelihood analysis using RAxML v.8 (33). For MrBayes, we used a mixed model, and for the maximum likelihood analysis, we used the generalized time reversible (GTR) model. Branch support was measured as the posterior probability of clades in the consensus tree for Bayesian analysis; and with 1000 bootstrapping replicates in the maximum likelihood analysis. To calculate the nonsynonymous ($d_N$) and synonymous ($d_s$) substitution rates between PriA and homologous subfamilies, we aligned all the sequences by codon using RevTrans 1.4 Server (*Wernersson and Pedersen, 2003*). To calculate the $d_N/d_s$ ratio we used codeml in the PAML four package (*Yang, 2007*). GC content, genome size, CDS content, and number of subsystems between the lineages were compared by using the T-test in the package R. All the boxplots were done with R.

The *A. oris* MG-1 strain (*Delisle et al., 1978*) was sequenced using an in-house Illumina MiSeq sequencing platform. We used Trimmomatic (*Bolger et al., 2014*) to filter the reads and Velvet v1.2.10 (*Zerbino and Birney, 2008*) to assemble the reads. The Whole Genome Shotgun (WGS) *A. oris* MG-1 project has been deposited at GenBank under the project accession [PRJNA327886].

## Metabolic model reconstruction and flux balance analysis

We applied the DOE Systems-biology Knowledgebase (KBase) to construct draft genome-scale metabolic models. The model reconstruction process was optimized as previously (*Satish Kumar et al., 2007*), and comprised of three steps: (i) genome annotation by RAST (*Aziz et al., 2008*); (ii) reconstruction of a draft model using the ModelSEED approach (*Henry et al., 2010*); and (iii) gapfilling of the model to permit growth and plug holes in mostly complete pathways (*Dreyfuss et al., 2013*). In the gap-filling process, we identified the minimal set of reactions that could be added to each model to permit biomass production in a medium containing every transportable metabolite. We also favored the addition of reactions that would permit more gene-associated reactions in each model to carry flux.

Once models were built, we applied flux balance analysis (FBA) (*Orth et al., 2010*) to predict minimal feasible media and classify reactions using a six step process: (i) set the biomass flux to a non-zero value; (ii) minimize the number of active exchange reactions to identify the minimal set of external nutrients that must be provided to permit growth; (iii) constrain exchange fluxes so that only the minimal exchanges are allowed to function; (iv) minimize and maximize each reaction flux to classify each reaction during growth on minimal media (*Mahadevan and Schilling, 2003*); (v) maximize biomass flux on minimal media and fix the biomass flux at its maximum value; and (vi) minimize the sum of all fluxes in the model to produce the simplest flux profile possible (e.g. removing all flux loops). Reactions with only positive or negative fluxes are classified as *essential*; reactions with only zero flux values are classified as *nonfunctional*; and reactions with zero and non-zero flux values are classified as *functional*.

For construction of the overall model per lineage, we identified all reactions that were associated with genes (i.e. not gapfilled) in at least 75% of the models included in the lineage, using a permissive e-value of 0.01. These reactions formed the basis of our lineage model. Then we applied the same gapfilling algorithm used with our genome models to permit the lineage model to grow. Finally, we applied our FBA pipeline to predict minimal media and classify reactions in the lineage model. All the models, associated genomes, minimal media predictions, reaction classifications, and flux predictions generated in this study are presented using the KBase Narrative Interface and are accessible at https://narrative.kbase.us/narrative/ws.17193.obj.1. See also *Figure 4—source data 1*, *Figure 4—source data 2* and *Figure 4—source data 3*.

## Biochemical analysis of PriA enzymes

The *priA* genes from Org15, Org10, and Org41 were synthesized by GeneArt (Thermo Fisher Scientific, USA). Additionally, *priA* genes from Org13, Org22, Org34, Org36, and Org39 were synthesized by GenScript (GenScript, USA). Codons were optimized for *E. coli* heterologous expression. The *priA* homologs from *A. oris* MG-1, *B. longum*, *B. gallicum* and *B. adolescentis* were PCR cloned from our genomic DNA collection. Oligonucleotide sequences of primers used in this study are included in *Supplementary file 3*. All genes were inserted into pET22b, pET28a (Novagen) for expression and protein purification, and pASK for complementation assays, by using the *Nde*I and *Hind*III restriction sites (*Noda-García et al., 2015*). The in vivo trpF and *hisA* complementation assays, and in vitro determination of the Michaelis-Menten steady-state enzyme kinetics parameters for both PRA and ProFAR as substrates, were done as previously (*Noda-García et al., 2013*; *Verduzco-Castro et al., 2016*; *Noda-García et al., 2010*). Lack of enzyme activity in vitro was confirmed using active-site saturation conditions, as before (*Noda-García et al., 2013*; *Verduzco-Castro et al., 2016*).

To create a *ΔpriA* mutation in *A. oris* MG1 1.5 Kbp fragments upstream and downstream of this organism were amplified by PCR (*Supplementary file 3*). The upstream fragment was digested with *Eco*RI and *Nde*I, the downstream fragment with *Nde*I and *Xba*I. The upstream and downstream fragments were ligated together in a single step. The fragment was cloned into pCWU3 precut with *Eco*RI and *Xba*I after digestion with appropriate enzymes. The generated plasmid was then introduced into *A. oris* MG-1 (Org21) by electroporation. Corresponding in-frame deletion mutants were selected by using mCherry fluorescence as a counter-selectable marker and resistance to kanamycin (*Wu and Ton-That, 2010*). The deletion mutant was confirmed by PCR and by sequencing of the entire genome of the resulting mutant strain.

## Crystallization, X-ray data collection, structure determination, and refinement

PriA_Org15 was expressed and produced in BL21 Magic cells bearing the plasmid pMCSG68_PriA_Org15. The protein was purified by immobilized metal-affinity chromatography (IMAC) followed by His6-tag cleavage using recombinant His-tagged TEV protease. A second IMAC step was used to remove the protease, the uncut protein, and the affinity tag. Concentrated protein (37 mg ml$^{-1}$) was crystallized by sitting-drop vapor-diffusion technique in 96-well CrystalQuick plates (Greiner Bio-One, USA). The crystals appeared at 289 K in conditions consisting of 0.2 M Li$_2$SO$_4$, 0.1 M CAPS:NaOH pH 10.5, and 1.2 M NaH$_2$PO$_4$/0.8 M K$_2$HPO$_4$. Prior to data collection crystals were cryoprotected in 2.4 M K$_2$HPO$_4$ and subsequently flash-cooled. Diffraction data were collected at 100 K. Native datasets were collected at 19-ID equipped with an ADSC quantum Q315r CCD detector at 0.979 Å wavelength. The images were processed by using the HKL3000 software suite (*Minor et al., 2006*). Molecular replacement was carried out by using the coordinates of PriA from *M. tuberculosis* (*Due et al., 2011*) used as a search probe in Phaser (*McCoy et al., 2007*). The initial model was then improved by the automatic rebuilding protocol in Arp/wArp, and further modified by iterations of manual rebuilding in COOT (*Emsley and Cowtan, 2004*) and fully anisotropic crystallographic refinement in PHENIX (*Adams et al., 2010*) with hydrogen atoms in riding positions. The PriA_Org15 model comprises residues Ser-2-Arg137 and Gly143-Ala247, 305 water molecules, 4 phosphate ions, and 1 CAPS moiety. The mFo-DFc difference map reveals two strong positive peaks (near Asp51 and Leu230) that could not be unambiguously assigned. The quality of the refined models was verified using the Molprobity server (*Chen et al., 2010*). Data collection statistics and the refinement results are provided as *Figure 7—source data 1*.

## Structural alignment, homology modeling and molecular docking

T-coffe package was used for all multiple sequence alignments (*Notredame et al., 2000*). Protein structural homology models of SubHisA_Org36 and SubTrpF_Org41 were based on the crystal structure of PriA from PriA_Org15 (PDB:4X2R; this study). A standard modeling strategy using Robetta and Rosetta 3.5 (47) was adopted. Molecular models of PRA and ProFAR were built using Molden (*Schaftenaar and Noordik, 2000*), and optimal atomic configuration of both substrates was obtained using Gaussian 09 (Gaussian Inc., Wallingford CT, USA) through a quantic geometry optimization using a self-consistent field at the Hartree-Fock 6–31G* level. Polar hydrogen atoms and Gasteiger-Marsili empirical atomic partial charges were added using AutoDockTools (*Morris et al., 2009*). An extensive configuration sampling of PRA and ProFAR binding biophysical interactions with PriA catalytic site was performed with Autodock Vina (*Trott and Olson, 2010*). Results were merged, refined, clustered, and energy sorted to produce a set of complex configuration predictions.

## Acknowledgements

We acknowledge Nelly Selem-Mójica, Víctor Villa-Moreno, José-Luis Steffani-Vallejo and Christian E. Martínez-Guerrero for bioinformatics support. We thank Sean Rovito and Angélica Cibrián-Jaramillo for useful comments and evolutionary discussions. We thank members of the Structural Biology Center at Argonne National Laboratory for data collection support. This work was supported by Conacyt Mexico, via grants 132376 to MCT and 179290 to FBG, as well as a scholarships to AJV. And by The National Institutes of Health, grant GM094585 to AJ, the US Department of Energy, under contract DE-AC02-06CH11357, and the National Science Foundation, grant 1611952 to CSH, and the National Institute of Dental and Craniofacial Research of the National Institutes of Health, grant DE017382, to HTT.

## Additional information

### Funding

| Funder | Grant reference number | Author |
| --- | --- | --- |
| Consejo Nacional de Ciencia y | 179290 | Ana Lilia Juárez-Vázquez |

| Tecnología | | Ernesto A Verduzco-Castro<br>Lianet Noda-García<br>Sofía Medina-Ruíz<br>Francisco Barona-Gómez |
|---|---|---|
| National Science Foundation | 1611952 | Janaka N Edirisinghe<br>Christopher S Henry |
| National Institutes of Health | GM094585 | Karolina Michalska<br>Gyorgy Babnigg<br>Michael Endres<br>Andrzej Joachimiak |
| National Institute of Dental and Craniofacial Research | DE017382 | Chenggang Wu<br>Hung Ton-That |
| Consejo Nacional de Ciencia y Tecnología | 132376 | Julián Santoyo-Flores<br>Mauricio Carrillo-Tripp |
| U.S. Department of Energy | DE-AC02-06CH11357 | Mauricio Carrillo-Tripp<br>Andrzej Joachimiak<br>Christopher S Henry<br>Francisco Barona-Gómez |

The funders had no role in study design, data collection and interpretation, or the decision to submit the work for publication.

### Author contributions

ALJ-V, Conceptualization, Formal analysis, Investigation, Methodology, Writing—original draft, Writing—review and editing; JNE, Data curation, Formal analysis, Investigation; EAV-C, Data curation, Formal analysis, Investigation, Methodology; KM, Data curation, Formal analysis, Investigation, Writing—original draft; CW, Investigation, Methodology; LN-G, SM-R, Conceptualization, Investigation, Methodology; GB, Formal analysis, Investigation, Project administration; ME, JS-F, Data curation, Methodology; MC-T, Formal analysis, Investigation, Methodology, Writing—original draft; HT-T, Supervision, Investigation, Methodology; AJ, Resources, Supervision, Funding acquisition, Investigation, Writing—original draft; CSH, Conceptualization, Data curation, Formal analysis, Supervision, Funding acquisition, Investigation, Methodology, Writing—original draft, Writing—review and editing; FB-G, Conceptualization, Resources, Data curation, Formal analysis, Supervision, Funding acquisition, Investigation, Methodology, Writing—original draft, Project administration, Writing—review and editing

### Author ORCIDs

Francisco Barona-Gómez, http://orcid.org/0000-0003-1492-9497

# Additional files

### Supplementary files

• Supplementary file 1. Genome analysis of the priA minus *Actinomyces oris* mutant.

• Supplementary file 2. Predicted affinities for PRA and ProFAR.

• Supplementary file 3. Primers used in this study.

### Major datasets

The following datasets were generated:

| Author(s) | Year | Dataset title | Dataset URL | Database, license, and accessibility information |
|---|---|---|---|---|
| Juárez-Vázquez AL, Barona-Gomez F | 2016 | The Whole Genome Shotgun (WGS) A. oris MG-1 project | https://www.ncbi.nlm.nih.gov/bioproject/PRJNA327886 | Publicly available at the NCBI genome database (accession no: PRJNA327886) |

| Michalska K, Verduzco-Castro EA, Babnigg G, Enders M, Barona-Gomez F, Joachimiak A | 2016 | Crystal structure of PriA from Actinomyces urogenitalis | http://www.rcsb.org/pdb/explore.do?structureId=4X2R | Publicly available at the RCSB Protein Data Bank (accession no: 4X2R) |

The following previously published dataset was used:

| Author(s) | Year | Dataset title | Dataset URL | Database, license, and accessibility information |
| --- | --- | --- | --- | --- |
| Juarez-Vazquez AL, Barona-Gomez F, Henry CS | 2016 | Actinomycetaceae Phylogenomics: Comparative Analysis of Models | https://narrative.kbase.us/narrative/ws.17193.obj.1 | Publicly available, subject to registering an account at KBase (accession no: ws.17193.obj.1) |

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
