## [Decision Letter]

Thank you for submitting your work entitled "Evolution of substrate specificity in a retained enzyme driven by gene loss" for further consideration at *eLife*. Your article has been favorably evaluated by Patricia Wittkopp (Senior Editor) and three reviewers, one of whom is a member of our Board of Reviewing Editors.

There are some issues that need to be addressed before acceptance, as outlined below:

The section on biochemical data requires a more clear version since there are questions about the significance and interpretation of the values reported. The interpretation of the parallel decay of both Trp and His pathways is not sufficiently convincing in the current version of the paper.

The interpretation of the mutations in the x-ray structure and models, together with the analysis of the MSA, are considered important but weak elements of the paper. The use of better methods and a more systematic approach will be required.

A new version should justify clearly the need of building complete metabolic models for the questions addressed in the paper.

*Reviewer #1:*

The basic proposal of the paper is interesting and well positioned. The study of PriA dual enzyme acting in two pathways (His and Trp) as a model of the changes in response to a large genomic delection of complete pathways, is an interesting and original idea.

The work presented include:

First the basic phylogenetic analysis of the protein family with the distribution of the aa pathways and the *PriA* gene. (The analysis of the larger family presented in the first figure seems irrelevant for the rest of the paper). The results are interesting and demonstrate the existence of a potential relation between the absence of the enzymes of the two pathways and the distribution of the *PriA* gene.

The importance of the pathways is analyzed with metabolic models of the different genomes. The exercise is interesting but it is debatable if it is really necessary for the conclusions obtained. ("Overall, we see from the modeling analysis that lineages I, III and IV are all undergoing the process of gene loss, with different genomes in these lineages losing different sets of metabolic genes." would not be enough with knowing that enzymes of the key His and Trp pathways are missing? What else do we learn that is important for the logic of the study?)

in vitro biochemical assays of some of the sequences of the key groups are presented to demonstrate the lack of restrictions of the enzymes of the SubHis and SubTrp groups. The results are partially consistent with the interpretation of the authors but the data suggest that the actual situation might be far more complex of what the model suggest. Indeed, it seems difficult to explain the poor catalytic efficiencies of the enzymes of the subHis and subTrp groups in the context of the adaptations to the corresponding missing pathways.

The analysis of the structure of the enzyme based on a single x-ray obtained for this paper and homology models of the other prototypic sequences. It is a correct analysis but very speculative. It provides very little solid evidence for the interpretation of the mutations.

The analysis of the variation at the sequence level based on the MSA alignmentof members of the family. In this case it is unclear how some specific positions are selected as key differences between SubTrp and SubHis. Some of the differences are not consistent with the conservations in the groups and it is unclear if they are conserved in other sequences of the family (for example, the T for S substitution in subHisA2_Org36 is not present in subHisA2_Org34).

Additionally, the analysis is not systematic and none of the various existing methods for the analysis of positions specific of subfamilies are used.

In summary, a very interesting idea and a potentially good model to explore the evolution of the function of an enzyme in similar genomes subject to specific gene losses. A great effort to combine different type of approaches from genomics to modelling, including biochemistry, phylogenetics and protein structures. To conclude: some not so optimal interpretation of the results and general lack of sufficiently convincing support for the proposed model.

*Reviewer #2:*

This Manuscript presents an imposing body of work showing how gene loss can drive the evolution of substrate specificity from retained enzymes. The authors did a thorough work, using multiple approaches to elaborate and support their hypotheses. More in detail:

1) Genome sequencing of the *A. oris* MG-1 strain.

2. Phylogenomic analysis of the Actinomycetaceae family, using 35 single-copy conserved proteins in 133 organisms.

3) Phylogenomic analysis 0f 205 single-copy proteins conserved in 41 organisms (33 from the Actinomycetaceae family, and 8 from the *Bifidobacterium* genus, used as an outgrup).

4) Metabolic model reconstruction and flux balance analysis on 33 Actinomycetaceae.

5) Molecular evolution analysis of the PriA gene within Actinomycetaceae.

6) Biochemical analysis of PriA enzymes.

7) Crystallographic analysis of PriA.

8) Structural modelling, analysis, and molecular docking to study the evolution of SubHisA2 and SubTrpF.

Overall, I found this work outstanding, and I think this is the kind of manuscript every evolutionary biologist would like to read. I really enjoyed it. I think the authors did a great job in designing the experiments, and in my opinion the methods, the results, the hypotheses and the conclusions are presented in a very clear and accurate form. I could not find any substantive concern about this work.

*Reviewer #3:*

This paper presents an interesting analysis of two phenomena that are under-studied: gene loss, and generalist-to-specialist evolution. In general, we know a lot on gene gain (by duplication or HGT) and relatively little on gene loss, and, everyone assumes that generalists evolving to specialists is a dominating trend in evolution, whereas it could may well be that the opposite trend, highlighted here, is equally common. Overall, the scope of the work from phylogenetic to crystal structures is impressive, and the evolutionary trend and unraveled mechanisms of gene loss are well supported.

On the critical side, I must admit that I'm not so convinced that the effects of specialization and general decay of activity can be distinguished in this case. By and large, both activities (Trp and His reactions) drop down and largely in parallel (Table 2). The detection limit for then in vitro assays is not specified (i.e., what n.d. values represent) but the residual, other activity may not be so far from the one detected and assigned as 'specialized'. The structural argumentation is weak – it's not that the authors did not make an effort to unravel the effects of mutations, but it's sometimes hard to understand what mutations are doing even within active-sites let alone in second-shell, and especially, when these enzyme variants are so crippled. In general, orthologous enzyme vary in activity, often over 3 orders, and even between relatively close species. So, I think that it's hard to say whether, and which part of, the change in activities can be attributed to gene loss and concomitant specialisation (to either Trp or His) and which part to an overall relief of selection (i.e., for both reactions). Finally, since these genome sequences are sporadic (unlike say hundreds of *E. coli* strains that were sequenced), some mutations may relate to drift (polymorphism) and may not represent a long-term "wild-type" sequence. Previous cases analysed by this group are far more clear-cut in this respect (the effect of specialization). These caveats should be explicitly addressed.

---

## [Author Response]

*The manuscript has been improved but there are some remaining issues that need to be addressed before acceptance, as outlined below:*

*The section on biochemical data requires a more clear version since there are questions about the significance and interpretation of the values reported. The interpretation of the parallel decay of both Trp and His pathways is not sufficiently convincing in the current version of the paper.*

Biochemical data. Reviewers questioned the detection limits of our enzyme assays, which were actually included in the original version, but we have now discussed them in relation to how the discovered enzymes have evolved. A key issue with regards to this discussion is the (preconceived) idea that we were reporting on a story related to specialization, as the generalist – specialist model prevails in the community, even when it is not referred to anywhere in the manuscript. After our results, we do not believe this model is appropriate when enzymes evolve under relaxation of purifying selection. We have clarified this point throughout the manuscript by explicitly stating “monofunctional, yet not necessarily specialized”. Indeed, the importance of our contribution is that it provides a comprehensive example that questions current models in many ways, and this was possible because we looked at natural – not directed or artificial – evolution. Also, as requested, we have also included a new figure to introduce the enzymes involved in this study, which will hopefully help the reader to better grasp the biochemical and evolutionary nature of the system we have investigated.

*The interpretation of the mutations in the x-ray structure and models, together with the analysis of the MSA, are considered important but weak elements of the paper. The use of better methods and a more systematic approach will be required.*

Structural biology data.Reviewers questioned if putative mutations identified using only one structure, together with our homology models/molecular dynamics simulations, could lead to the conclusions reached. Indeed, reviewers are right and certainly our structural evidence is only useful in the context of other biochemical and evolutionary data. We did actually try to do some further sequence analysis, as suggested by you, but this was not possible due to the little data available. As reviewer 3 states “it’s not that the authors did not make an effort to unravel the effects of mutations, but it’s sometimes hard [...] when these enzymes are so crippled [...] (and) these genome sequences are sporadic”. In other words, there is nothing else we could do with the available data. Along these lines, it should be noted that sequence conservation under purifying selection is an assumption that lacks evidence – again, a preconceived notion as in previous point. In the light of this recognition, we proceeded more carefully about any conclusion related to the structural data by modifying all related writing throughout the manuscript. Specifically, we explicitly say this in a paragraph included at the end of the structural sub-section. We believe this explains our limitations but also embraces some of the specific possibilities raised by the reviewers, without diminishing our structural work.

*A new version should justify clearly the need of building complete metabolic models for the questions addressed in the paper.*

Metabolic modeling data. Reviewers questioned the use of this approach, which hopefully is better justified now that we have provided a better organization of the relevant sections. This includes re-doing of Figure 2 (now Figure 3), and re-allocation of some of the data included in this figure, as Figure 4—figure supplement 1 to the Figure reporting the metabolic modeling, originally 3 now 4. We also worked on the writing when appropriate, to make clear that although occurrence / absence of any given gene can be noted relatively straightaway, overreaching and risky conclusions can only be avoided if genomes are seen through the eyes of high-quality annotations, such as metabolic models. We argue that this should become a new standard when assuming that enzymes are missing or present, which in turn translate into specific hypothesis.